# Protein structural features predict responsiveness to pharmacological chaperone treatment for three lysosomal storage disorders

Jaie Woodard[1], Wei Zheng[1], Yang Zhang[1,2]*

**1** Department of Computational Medicine and Bioinformatics, University of Michigan, Ann Arbor, Michigan, United States of America, **2** Department of Biological Chemistry, University of Michigan, Ann Arbor, Michigan, United States of America

* zhng@umich.edu

**Data Availability Statement:** All relevant data are within the manuscript and its Supporting Information files.

## Abstract

Three-dimensional structures of proteins can provide important clues into the efficacy of personalized treatment. We perform a structural analysis of variants within three inherited lysosomal storage disorders, comparing variants responsive to pharmacological chaperone treatment to those unresponsive to such treatment. We find that predicted ΔΔG of mutation is higher on average for variants unresponsive to treatment, in the case of datasets for both Fabry disease and Pompe disease, in line with previous findings. Using both a single decision tree and an advanced machine learning approach based on the larger Fabry dataset, we correctly predict responsiveness of three Gaucher disease variants, and we provide predictions for untested variants. Many variants are predicted to be responsive to treatment, suggesting that drug-based treatments may be effective for a number of variants in Gaucher disease. In our analysis, we observe dependence on a topological feature reporting on contact arrangements which is likely connected to the order of folding of protein residues, and we provide a potential justification for this observation based on steady-state cellular kinetics.

## Author summary

Pharmacological chaperones are small molecule drugs that bind to proteins to help stabilize the folded state. One set of diseases for which this treatment has been effective is the lysosomal storage disorders, which are caused by defective lysosomal enzymes. However, not all genotypes are equally responsive to treatment. For instance, missense mutants that are particularly destabilized relative to WT are less likely to respond. The availability of datasets containing responsiveness data for large numbers of mutants, along with crystal structures of the protein involved in each disease, make machine learning methods incorporating sequence-based and structural data feasible. We hypothesize that data from two diseases, Fabry and Pompe disease, may be useful for predicting responsiveness of variants

**Funding:** This work is supported (awards to YZ) in part by the National Institute of General Medical Sciences (GM136422, S10OD026825), the National Institute of Allergy and Infectious Diseases (AI134678), and the National Science Foundation (IIS1901191, DBI2030790, MTM2025426). The funders had no role in study design, data collection and analysis, decision to publish, or preparation of the manuscript.

**Competing interests:** The authors have declared that no competing interests exist.

in the related Gaucher disease. Results suggest that many rare variants in Gaucher disease could be amenable to existing drugs. Results also suggest that drug responsiveness depends on protein topology in such a way that mutations in early-to-fold residues are more likely to be non-responsive to pharmacological chaperone treatment, which is consistent with a simple kinetic model of stability rescue. This study provides an example of how machine learning can be used to inform further studies towards personalized treatment in medicine.

## Introduction

One mechanism by which a missense mutation can exert a pathogenic effect is by destabilizing the protein in which it is located, leading to deficiency of the protein, not just its enzymatic activity [1–5]. A smaller amount of the protein is then available to carry out its function, particularly since the destabilized protein is more likely to be cleared by quality control machinery [6–8]. One potential treatment for diseases caused by destabilized variants is a small molecule drug that helps stabilize the mutant protein, called a pharmacological chaperone [9–13]. Pharmacological chaperones have become popular as potential treatments for several diseases, including some lysosomal storage disorders, over the past 20 years, and the drug Migalastat (with the commercial name Galafold) based on this approach, has been approved for treatment of Fabry disease [14–16]. The drug binds the protein as an inhibitor, assisting folding in the Endoplasmic Reticulum. This follows an initial observation that the product and inhibitor galactose, of which Migalastat is an analog, improves activity and patient condition, although it is not a feasible therapeutic due to the high required dose [9,17]. Non-inhibitory chaperones have also been proposed and tested at various stages of the approval process [12,18–20], which may overcome dosing complications introduced by inhibitory effects. For lysosomal storage disorders of appropriate genotype, oral pharmacological chaperone treatment can replace or supplement intravenous enzyme replacement treatment, which is expensive and does not cross the blood-brain barrier.

The lysosomal storage disorders are a set of over 50 rare genetic diseases characterized by defective processing and build-up of substrates in the lysosome, often due to mutation in a lysosomal enzyme [21,22], which include Gaucher, Fabry, and Pompe disease. Gaucher disease [23] is the most common lysosomal storage disorder, consisting of three subtypes, the rarer two of which include neurological symptoms. Pharmacological chaperone treatment has shown promise for treatment of the disease [23,24–28], although no pharmacological chaperone drugs are currently FDA approved for this purpose.

Some mutants of a given protein are more responsive to pharmacological chaperone treatment than others. For Fabry disease, it has been shown that residues in the active site or those that grossly destabilize the protein are less likely to be responsive to treatment [16]. An interesting biophysical and bioinformatic question is then whether responsiveness to treatment can be predicted from molecular properties, utilizing machine learning methods. Previous studies have demonstrated success in predicting pathogenic variants and/or responsiveness to pharmacological chaperone treatment for Fabry disease [29–31]. Here, we seek, in particular, to elucidate how protein structure can inform upon whether a mutation is responsive to treatment. Protein structures have adopted an increasingly important role in many aspects of genetics, *e.g.*, [32–35]. We hypothesized that pathogenicity would depend with reasonably high prediction accuracy on aspects obtainable from structure and computation, such as the

predicted stability change and local topology, and that predictions may be transferrable between proteins that cause differing lysosomal storage disorders.

Cell-based assays can effectively predict response to pharmacological chaperone treatment. We analyze datasets for the cases of Fabry and Pompe disease [15,36], for which large numbers of variants have been tested. We find that free energy change $\Delta\Delta G$ upon mutation predicted by our program EvoEF [37] is predictive of responsiveness to treatment, with particularly destabilized variants more likely to be unresponsive. Furthermore, we find a dependence on a topological measure which predicts that mutations in residues early to fold will be less likely to be responsive to pharmacological chaperone treatment, which we rationalize according to cellular folding kinetics. We train machine learning models on the two datasets, using a combination of sequence-based and structural features, and we use the resulting models to predict responsiveness for Gaucher disease variants. We find that many variants of Gaucher disease are predicted to be responsive to pharmacological chaperone treatment, using tested variants as a point of comparison.

# Results

## Key structural features of missense mutations

A number of structural features were considered for each of the three proteins associated with lysosomal storage disorder that were analyzed in this study, as illustrated in Fig 1. First, $\Delta\Delta G$ was estimated using the program EvoEF [37]. Consistent with previous results [16], mutations not responsive to pharmacological chaperone treatment showed a statistically significant increase in mean predicted $\Delta\Delta G$ of the distribution, for both Fabry and Pompe disease proteins (Fig 2). Other information we considered in decision tree analysis was the number of residues in contact with the mutated residue (6 or more heavy atom pairwise contacts within 8 Angstroms), whether the mutated residue was within 5 Angstroms of the ligand in the crystal structure, the crystallographic B-factor of the alpha carbon, and biomolecular circuit topology information based on residue-residue contacts.

Circuit topology provides a way to formalize the relationships between pairs of contacts within a protein or other linear polymer. The interval of a contact is defined as the span of

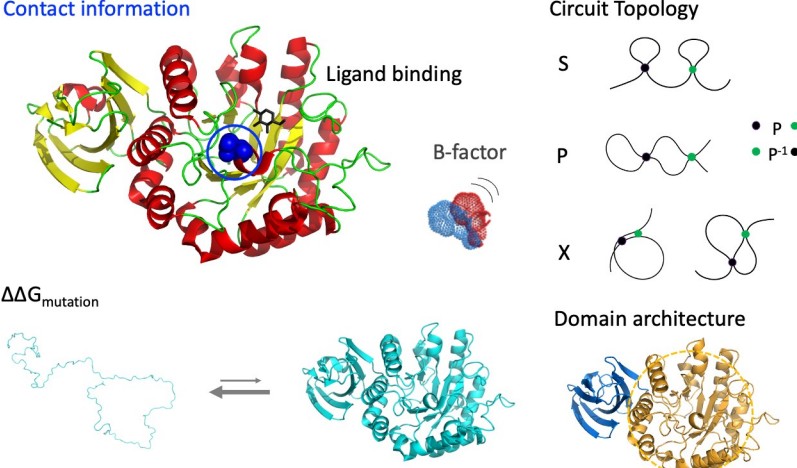

**Fig 1. Illustration of structural features considered in machine learning.** Features considered were the number of residues in contact with the mutated residue, whether the residue is in contact with the ligand at the catalytic site, the crystallographic B-factor or temperature factor, the change in free energy upon mutation, the "local" circuit topology of the residue, and whether the residue is located in the catalytic domain.

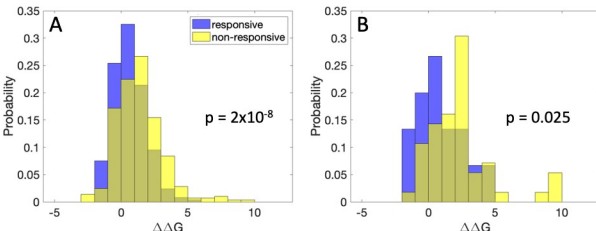

**Fig 2. Stability change and domain location features of treatment responsive and non-responsive variants.** Shown are stability histograms, comparing variants responsive to treatment (blue) and non-responsive to treatment (yellow). (A) $\Delta\Delta G$ for Fabry disease (kcal/mol). (B) $\Delta\Delta G$ for Pompe disease (kcal/mol).

sequence between the two contacting residues, along the protein chain. Two contacts may be in three types of relations: series, parallel, or cross, as illustrated in Fig 1 and described in previous publications [38–40]. In particular, the parallel relation, relevant to this study, occurs when one interval is contained entirely within the other. A distinction is made between strict parallel and inverse parallel. One contact is said to be in parallel with another contact if it is contained in its interval, while one contact is said to be in inverse parallel with another contact if it contains its interval (i.e., the outer contact in a parallel relation). We then define local circuit topology to be the number of contacts with a particular relation relative to any contact formed by the mutated residue, consistent with (*manuscript in preparation*).

The domain architecture of each of the three proteins was considered visually in further detail (S1 Fig). Here, the central residue numbers of secondary structural elements (alpha helices and beta strands) which interact in three-dimensional space are connected by a curve. Interestingly, the non-catalytic domain in the case of glucocerebrosidase (Gaucher) consists of intertwined N- and C-terminal regions of the chain, leading to a large number of contacts in parallel. It is important to consider that differences in domain structure may affect transferability of mutational effects.

## Decision tree analysis and predictions

Pharmacological chaperone responsiveness, based on enzymatic activity in cellular assays, as described in referenced publications [15,36], was predicted in this study. A decision tree based on the Fabry dataset (Fig 3A) was generated using R-part in R, using the default algorithm, with the complexity parameter determining the number of decision nodes chosen to first optimize the MCC of predictions on the Pompe dataset and then, for equivalent values on the Pompe dataset, to locally optimize the MCC of the Fabry dataset (S1 Table). The first branching is according to predicted $\Delta\Delta G$ of the mutation, where destabilization greater than 1.7 kcal/mol predicts non-responsiveness to pharmacological chaperone treatment. Next, if the mutated residue is in contact with the ligand in the crystal structure, non-responsiveness is also predicted. Otherwise, if the number of contacts is sufficiently small, the mutation is predicted as responsive. For greater numbers of contacts, the tree again branches at $\Delta\Delta G$, where values smaller than 0.94 but greater than -1.4 kcal/mol are predicted as responsive. The most stabilizing mutations are predicted to be non-responsive. For values of $\Delta\Delta G$ greater than 0.94 (but still less than 1.7 kcal/mol), the number of inverse parallel relations determines responsiveness, where greater numbers of inverse parallel relations predict non-responsiveness.

The decision tree correctly predicts responsiveness or non-responsiveness for three Gaucher disease mutations to the chemical chaperone N-nonyl-deoxynojirimycin in a cell-based assay (Table 1). Specifically, N370S and G202R are predicted to be responsive, while L444P is predicted to be non-responsive, consistent with known mutational effects [24].

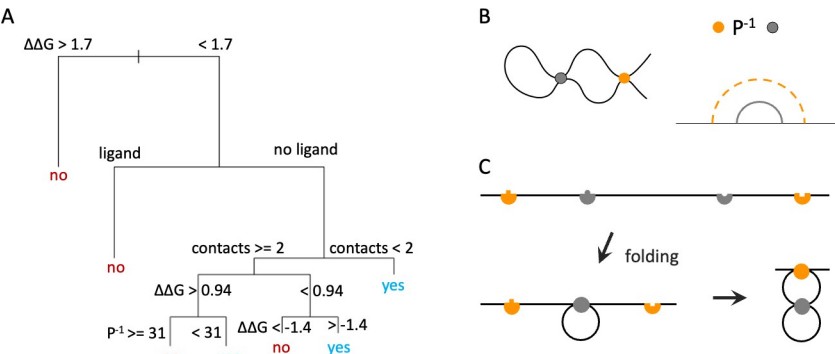

**Fig 3. Decision tree of structural features and importance of the inverse parallel relation.** (A) Decision tree constructed in R-part from the Fabry dataset, MCC = 0.39 on Fabry dataset, MCC = 0.49 on Pompe dataset. Tree predicts whether the mutation is responsive (yes) or non-responsive (no) to pharmacological chaperone treatment. Units of ΔΔG are kcal/mol. (B) The orange contact is in inverse parallel ($P^{-1}$) relation with the gray contact. An example curve satisfying this condition (left) and topology diagram for this relation (right) are shown. (C) Model of folding of the parallel relation, with the gray contact forming first, facilitating formation of the orange contact. The top diagram shows the contacting points of the unfolded chain, with complementary points in the same color; bottom diagrams show a partially folded chain with the inner, gray contact formed (left) and a folded chain with both contacts formed (right).

Table 2 shows several structural properties of these mutations. Notably, the non-responsive mutation L444P is predicted to be the most destabilizing of the first three mutations, above the 1.7 kcal/mol threshold of the decision tree. (Additional table columns are explained in later sections of the text.) The tree is less effective on predicting the responsiveness of additional variants to Ambroxol [41,42], correctly predicting only N188S (last four mutations of Table 1). Although non-responsive to N-nonyl-deoxynojirimycin, L444P was shown to be responsive in other cases [43,44].

An important aspect of the decision tree is the branching according to the number of inverse parallel relations, for intermediate ΔΔG values. The inverse parallel relation and implications for folding are depicted in Fig 3B and 3C. Considering intervals between contacting points, the contact in inverse parallel relation (orange) is outside of the first contact (gray, Fig 3B). Formation of the gray contact will then shorten the distance along the chain of the contacting points of the orange contact (Fig 3C). The gray contact is then expected to form first during the folding process, with the contact in inverse parallel folding later on.

To verify that residues with a large number of inverse parallel relations tend to be early to fold, we referenced hydrogen-deuterium exchange mass spectrometry experiments for a TIM barrel protein [45]. We found that residues in regions conferring the strongest protection

**Table 1. Predicted responsiveness of Gaucher disease mutations.**

|        | Observed responsive | Single tree responsive | Fabry ML responsive (p1) | Pompe ML Responsive (p1) |
|--------|---------------------|------------------------|--------------------------|--------------------------|
| N370S  | yes                 | yes (0.60)             | yes (0.41)               | yes (0.44)               |
| L444P  | no*                 | no (0.28)              | no (0.24)                | yes (0.42)               |
| G202R  | yes                 | yes (0.60)             | yes (0.36)               | yes (0.60)               |
| F213I  | yes                 | no (0.34)              | yes (0.62)               | no (0.30)                |
| R120W  | yes                 | no (0.28)              | no (0.05)                | no (0.24)                |
| N188S  | yes                 | yes (0.60)             | yes (0.66)               | no (0.31)                |
| D409H  | no                  | yes (0.60)             | yes (0.38)               | yes (0.42)               |

*Although non-responsive to N-nonyl-deoxynojirimycin, L444P was shown to be responsive in other cases [43,44].

**Table 2. Structural properties of Gaucher disease mutations.**

|  | EvoEF ΔΔG (kcal/mol) | Binds ligand | Number of contacts | Catalytic domain | P$^{-1}$ relations |
|---|---|---|---|---|---|
| N370S | 0.28 | No | 3 | Yes | 183 |
| L444P | 1.96 | no | 2 | No | 43 |
| G202R | -1.00 | No | 2 | Yes | 219 |
| F213I | 1.05 | No | 5 | Yes | 219 |
| R120W | 2.18 | No | 10 | Yes | 145 |
| N188S | 0.42 | No | 2 | Yes | 208 |
| D409H | 0.55 | No | 2 | Yes | 79 |

from exchange (first to fold) have a higher number of inverse parallel relations on average, followed by residues exhibiting intermediate protection, followed by other residues (S2 Fig). This suggests that residues that are earlier to fold tend to have a larger number of inverse parallel relations for the TIM barrel fold, which is found in the proteins associated with lysosomal storage disorders referenced in this study.

Glycine to arginine is a severe mutation in terms of change in charge, size, and torsional propensity. It is one of the most common pathogenic mutations and has a strong bias towards pathogenicity, according to the ADDRESS database of over 20K pathogenic and benign variants [3]. However, the position of glycine 202 within a loop in the structure of glucocerebrosidase prevents the mutated residue from strongly clashing with other residues in the protein, and the predicted ΔΔG is in fact negative. In the ADDRESS dataset, benign glycine to arginine mutations are shifted towards more negative ΔΔG relative to pathogenic ones (Fig 4A), and the probability of pathogenicity for a stabilizing mutation is 0.69 vs. 0.91 for a destabilizing mutation, indicating that the small ΔΔG value for G202R causes the mutation to more likely be benign. Notably, G to R mutations exhibit a wide range of ΔΔG values, with many predicted highly destabilizing (predominantly pathogenic) mutations, as well as many predicted stabilizing (predominantly benign) ones. The EvoEF optimized structure predicts in fact that G202R is highly solvent exposed. Interestingly, two arginine residues, including the mutated residue, are in close vicinity, such that ΔΔG may be more positive than predicted.

## Kinetic model of folding in the Endoplasmic Reticulum

To establish whether the dependence of drug responsiveness on the number of inverse parallel relations could be explained by kinetics of the folding and interaction system, we constructed a simple kinetic model. Referencing Fig 3B and 3C, residues that fold early in the folding process are expected to have a large number of contacts in inverse parallel relation. According to the two-state folding model [46,47], mutation of such early folding residues is expected to change

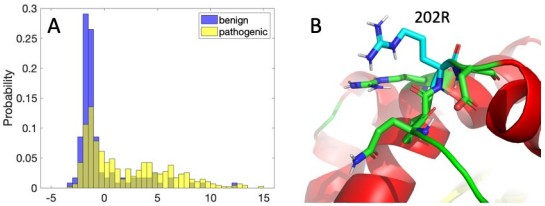

**Fig 4. Glycine to arginine mutation statistics and predicted structure.** (A) Histograms of ΔΔG (kcal/mol) from the ADDRESS database for glycine to arginine mutations. T-test p-value = 2x10$^{-10}$. (B) EvoEF optimized structure for the G202R mutation (using the BuildMutant command within EvoEF).

the folding rate, rather than the unfolding rate. Intuitively, we might expect that mutations that speed the unfolding rate (*i.e.*, with few inverse parallel relations) could be more easily rescuable by stabilizing molecules that bind to the folded state. This would be consistent with the results of the decision tree in Fig 3, which predicts responsiveness for few numbers of parallel relations but non-responsiveness for large numbers of parallel relations.

A kinetic model quantifying this argument is shown in Fig 5. Unfolded protein, U, is produced on polyribosomes and enters the Endoplasmic Reticulum (ER). The protein folds to state F with rate $k_f$, or is degraded at rate $k_p$. From the folded state, the protein may unfold back to state U with rate $k_u$. It may also bind the pharmacological chaperone ligand with rate $k_{on}L$, or it may be exported from the ER with rate $k_{out}$ to state E. From the chaperone-bound state $F_B$, the protein may dissociate from the chaperone and re-enter state F, or it may also be exported to state E. Steady-state results for parameter values given in the methods section, according to Eqs 1–3, are shown in Fig 5B, for mutations destabilizing the protein by speeding the unfolding rate (blue, cyan) or slowing the folding rate (red, magenta). Mutations that speed the unfolding rate are seen to be more responsive to treatment for a given ΔΔG value. While in actuality, mutations are likely to have fractional phi values, affecting both folding and unfolding rates to some extent, and the protein is likely to have multi-state folding properties, this model suggests that the dependence on number of inverse parallel relations may be explainable by protein folding propensities. While it is possible that additional biophysical factors contribute to the trend of circuit-topology-dependent responsiveness, it is interesting that both a kinetic model based on a reasonable model of enzyme supply to the lysosome and the observation that large numbers of inverse parallel relations decrease the chance of responsiveness are consistent with a picture in which speeding of the unfolding process is more likely to be rescued by pharmacological chaperone treatment than slowing of the folding process, for equivalent ΔΔG.

$$F = \frac{k_f P}{k_p + k_f}\left(k_u + k_{on} + k_{out} - \frac{k_f k_u}{k_p + k_f} - \frac{k_{off} k_{on}}{k_{off} + k_{out}}\right) \qquad (1)$$

$$F_B = \frac{k_{on} F}{k_{off} + k_{out}} \qquad (2)$$

$$\frac{dE}{dt} = k_{out}(F + F_B) \qquad (3)$$

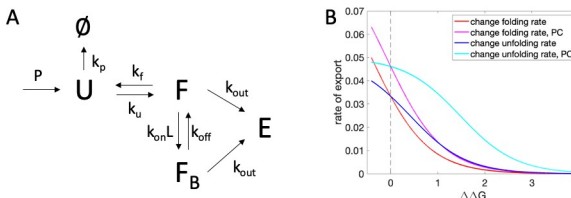

**Fig 5. Kinetic model of ER folding and export, with and without pharmacological chaperone (PC).** (A) The kinetic model, including unfolded state, U, folded state F, folded state bound to chaperone $F_B$, and protein exported from the ER E. P is the rate of production of unfolded protein, $k_f$ is the folding rate, $k_u$ is the unfolding rate, $k_{on}L$ is the rate of chaperone binding, $k_{off}$ is the rate of dissociation of protein and chaperone, and $k_{out}$ is the rate of export. (B) Steady state solutions for rate of export, $\frac{dE}{dt}$, for mutations affecting the folding or unfolding rate, with or without pharmacological chaperone (PC) treatment.

## Machine learning of responsiveness to pharmacological chaperone treatment

Machine learning was carried out using the Auto-ML capability of h2o. For the Fabry dataset, the best performing model was an ensemble of methods, while for the Pompe dataset it was an XGBoost method (S2 Table). Cross-validation AUC was high in both cases: 0.829 and 0.881, respectively (Table 3). The MCC on the same dataset (assessing performance on the cross-validated data of the entire collection of mutations) was 0.839 for Fabry and 0.712 for Pompe. In addition to cross validation on the same dataset, we obtained the MCC using the other dataset as the test set. (i.e., training on the Fabry dataset and testing non the Pompe dataset, or training on the Pompe dataset and testing on the Fabry dataset). The MCC for the performance of the Fabry-trained model on the Pompe dataset was 0.609, while the MCC of the Pompe-trained dataset on the Fabry data is substantially lower (0.282), likely because of a smaller amount of training data that does not encompass the full feature space of the Fabry dataset.

We predicted drug responsiveness for Gaucher disease variants based on the other two datasets. 105 of 203 mutations were predicted by both methods to be responsive to treatment, and 47 were predicted as non-responsive by both methods, with an MCC comparing the predictions of the two methods on the Gaucher dataset of 0.455. The prevalent mutation N370S was predicted to be responsive to treatment by both the Fabry and Pompe dataset based methods, as was G202R. L444P, which is known to be more difficult to treat via pharmacological chaperones [24], was predicted as non-responsive by the Fabry based method and responsive by the Pompe based method (although it had the lowest p1 of the three mutations, see Table 1). Complete predictions are given in S3 Table.

In general, based on our statistics and properties of the training data and models, we might expect the machine learning (H2O) results based on the Fabry dataset to yield the highest accuracy results. We in fact see better performance of the Fabry machine learning model on the additional Ambroxol responsiveness variants (last four mutations, Table 1). Of the two mutations incorrectly predicted, D409H had a relatively low p1 value. R120W, however, was shown experimentally to be responsive to Ambroxol despite a very low p1. Table 2 shows that this variant has several properties predicted to increase the propensity to be non-responsive to treatment, including a ΔΔG of greater than 1.7 kcal/mol and a large number of contacts involving the mutated residue. It could be informative to study this mutant in further detail using computational and/or experimental methods. The larger dataset of Fabry disease, combined with the sophisticated machine learning method, yields increased performance over the other methods described.

A molecular dynamics study [29] reported that glycine residues in particular had lower residual activity than might be expected based on RMSF, in relation to other residues. Glycine in general may be expected to have high flexibility, due to the large amount of torsional accessibility in the Ramachandran map relative to other residues; in addition, referencing our work summarizing mutations in 70,597 proteins [3], and as also seen previously [48], mutations from glycine to other residues tend to be pathogenic more often than mutations from most other residues (see S3 Fig), which could be a reason for the observed anomalous behavior of glycine mutations.

**Table 3. Results of cross validation incorporating H2O Auto-ML.**

| Dataset | AUC same | MCC same | MCC other |
|---------|----------|----------|-----------|
| Fabry | 0.829 | 0.734 | 0.609 |
| Pompe | 0.881 | 0.712 | 0.282 |

## Discussion

Pharmacological chaperone treatment has been explored in the case of several diseases [9,10,21], with much of the focus centering on certain lysosomal storage disorders. For such disorders, the small molecule chaperone helps to stabilize destabilized but still potentially enzymatically active variants of an enzyme. In drug-amenable proteins, key residues for catalytic activity are intact, while the variants affect the stability of folding. Prediction of which variants are amenable to such treatment has similarities to the prediction of pathogenicity, and factors such as excessive destabilization and ligand-contact, which contribute towards pathogenicity, tend also to contribute towards lack of rescue. However, the range of these values across variants is intrinsically shortened in the case of chaperone responsiveness, since all mutations are pathogenic.

It was suggested that the L444P variant was non-responsive to treatment due to its location in a domain other than the one containing the active site which is bound by the drug [24]. However, referencing the Fabry dataset [15], we see that mutations in residues not contained in the active site domain are often responsive to pharmacological chaperone treatment. The leucine to proline mutation type generally has a strong bias towards pathogenicity [3]. Therefore, the mutation type and stability change of the L444P mutation may be the source of pathogenicity, rather than (or in addition to) domain location. Considering these factors, it may be useful to test for responsiveness of other mutants in the non-catalytic domain of glucocerebrosidase. For instance, R463C, which comes in contact with the catalytic domain, is predicted to be responsive to treatment.

We note that our machine learning approach does not explicitly account for amino acid change propensities in ADDRESS [3]; these may be considered in addition to the given predictions. For instance, although arginine to cysteine mutations are most often pathogenic, these mutations are common and are benign in many cases. In fact, in the case of Fabry disease, several arginine to cysteine mutations show residual basal activity and responsiveness to pharmacological chaperone treatment [13]. Since similar propensities that cause a mutation to be pathogenic also cause it to be non-responsive to treatment (e.g., strong destabilization), a transfer learning approach, utilizing data on pathogenicity may be applicable.

It is interesting that the G202R mutation, which is known to be rescuable by pharmacological chaperone treatment, is predicted to stabilize, rather than destabilize the protein. Referencing Fig 2, this is the case for many variants. It is possible that some variants detrimentally affect folding without increasing the locally computed $\Delta\Delta G$. For instance, a mutation may make the protein more prone to misfold upon global unfolding. The bound drug would decrease the propensity for unfolding, reducing the propensity for misfolding and degradation. Kinetics may also be important [6,49]. A mutation that approximately equally slows unfolding and slows folding may have a pathogenic effect, for instance, if there is a particular importance of folding (vs. unfolding) rate. Although variants that are predicted to be especially stabilized are more likely to be non-responsive to treatment, the case of enzymes with $\Delta\Delta G$ predicted as sufficiently stabilizing to drive the prediction of non-responsiveness is fairly rare across clinically observed disease-causing mutations.

Our results support the key biological insight that kinetics of folding and binding in the cellular environment plays an important role in determining the responsiveness of variants to pharmacological chaperone treatment in the case of lysosomal storage disorders. Kinetic information is contained in circuit topology features derived from the crystal structure; contacts with many inverse parallel relations are assumed to be early to fold. According to a kinetic model built to approximate folding and drug binding in the cell, early to fold contacts are more difficult to rescue. This is consistent with the observation that mutants of residues with a large number of inverse parallel relations are statistically less likely to be responsive to pharmacological chaperone treatment. These factors link cellular kinetics and protein topology and demonstrate that both are likely important in determining drug responsiveness in this system.

It will be important to assess how results of molecular dynamics [29,50] and monte carlo simulations (e.g., RMSF or number of native contacts at point of residue unfolding) correlate with residue-based circuit topology. In the case that they are somewhat redundant, circuit topology measures might be a quick alternative to more computationally intensive simulations, suitable for analysis of large numbers of proteins at once, and in the case that they provide unique information, they might serve complementary roles in structure-based analysis. Our results and models suggest that different systems might depend differently on the numbers of relations of a given type, depending on details of the kinetics of the system.

It will be interesting to investigate the transferability of information from these lysosomal storage disorders, along with larger amounts of information on pathogenicity of human variants, to the proteins involved in other pharmacological-chaperone-amenable diseases. Predictions on responsiveness of variants can be used when considering whether a treatment may be effective, as well as for gauging the difficulty of the target when pursuing drug design. While we have utilized crystallographic structures in this study, NMR structures and computationally predicted structures and ligand binding sites may also be utilized in future studies of other proteins. Ultimately, protein-centric methods can be extended to other disease types, including cancers, with the aim of improving precision medicine based treatment decisions and facilitating drug discovery.

## Methods

### Datasets and structures

Existing datasets were referenced for response to pharmacological chaperones for Fabry [15] and Pompe [36] diseases. A list of Gaucher disease variants was obtained from a published article [23]. Crystal structures 2V3D (Gaucher [51]), 3GXT (Fabry [52]), and 5NN5 (Pompe [53]) were referenced.

### Domain divisions

For 2V3D and 3GXT, Pfam was referenced to determine domain divisions, and the catalytic domain was identified by visual inspection. For 5NN5, Pfam incorrectly determined domain divisions. This structure was aligned to the first two structures using TM-align [54], and domain divisions were determined according to the alignment. The ending residue of the catalytic domain was different for alignment to the two structures, and the average sequence position was chosen as the ending point.

### Protein free energy change upon mutation

Change in free energy upon mutation, $\Delta\Delta G$, was estimated using the program EvoEF [37]. The mutant protein was built and refined in EvoEF, following refinement of the WT protein and subsequent calculation of reference free energy for comparison.

### Contact information

Number of residues in contact with the mutated residue was calculated for each residue type by considering the number of residues forming greater than or equal to five atom-atom pairwise contacts of any type within 4.5 Angstroms. Circuit topology relations were calculated using existing code and methods [38]. For each residue, the number of relations of a given type with respect to any contact formed by the mutated residue was considered as "local" circuit topology for that residue.

## PSSM generation

To construct the PSSM, multiple sequence alignments (MSAs) for the query sequence were first generated by DeepMSA2, which utilizes HHblits [55,56], Jackhmmer [57] and HMMsearch [58] to iteratively search two whole-genome sequence databases (Uniclust30 [59] and UniRef90 [60]) and four metagenome sequence databases (Metaclust [61], BFD [62], Mgnify [63], and IMG/M [64]). DeepMSA2 contains three approaches, dMSA, qMSA, and mMSA, where the dMSA pipeline is equivalent to the DeepMSA [65] pipeline.

For the dMSA pipeline, HHblits2, Jackhmmer and HMMsearch were used to search the query against Uniclust30 (version 2017_04), UniRef90 and Metaclust, respectively. In Stage 2 and 3 of dMSA, homologs identified by Jackhmmer and HMMsearch, respectively, were gathered into a custom HHblits format database, which would be searched by HHblits2 using the MSA input from the previous stage to generate new MSAs. As an extension of dMSA, qMSA (which stands for "quadruple MSA") performs HHblits2, Jackhmmer, HHblits3, and HMMsearch searches against Uniclust30 (version 2020_01), UniRef90, BFD, and Mgnify, respectively, in four stages. Similar to dMSA Stage 2 and 3, the sequence hits from Jackhmmer, HHblits3 and HMMsearch in Stage 2, 3 and 4 of qMSA were converted into HHblits format database, against which the HHblits2 search based on MSA input from the previous stage is performed. In mMSA (or "multi-level MSA"), the qMSA Stage 3 alignment was used as a probe by HMMsearch to search through the IMG/M database and the resulting sequence hits were converted into a sequence database. This mMSA database was then used as the target database, which was searched by HHblits2 with three seed MSAs (MSA from dMSA stage 2, qMSA stage 2 and 3), to derive three new MSAs. These steps resulted in 10 MSAs in total (i.e., 3 from dMSA, 4 from qMSA, and 3 from mMSA), which were scored by TripletRes [66] contact prediction, where the MSA with the highest probabilities for top 10L (L is the sequence length) all range contacts ($C\beta$-$C\beta$ distances<8Å) were be selected.

The final MSA from DeepMSA2 is formatted as "a3m" file. This "a3m" format file was first converted to Blast [67] format database. Then, the query sequence was used to search this database, using PSI-Blast, and finally the PSSM file was generated by PSI-Blast.

## Kinetic model

Equations for the kinetic model are given in the main text. Kinetic parameters were chosen as: $P = 0.1$, $k_p = 10$, $k_{on}L = 0.1$ (or 0.00001 without chaperone), $k_{off} = 0.01$ (or 0.00001 without chaperone), $k_{out} = 0.01$, $k_u = 0.01$ or ranging from 0.005 to 10 in increments of 0.0005, $k_f = 10$ or ranging from 0.01 to 20 in increments of 0.001, $\Delta\Delta G = 4.0963–0.593^*\ln(k_f/k_u)$.

## Machine learning

The machine learning feature set consisted of the type of residues mutated from and to (employing one-hot encoding), the number of residues of each type in contact with the mutated residue, the crystallographic B-factor of the alpha carbon, the EvoEF-predicted $\Delta\Delta G$ of mutation, whether each residue was within 5 Angstroms of the active site ligand, and the deep-Multiple-Sequence-Alignment PSSM data for a given residue.

AutoML was performed using the program h2o, with 10-fold cross validation, using default options, and a maximum runtime of 500 seconds.

## Supporting information

**S1 Fig. Diagrams of proteins with mutants associated with lysosomal storage disorders.**
(PDF)

**S2 Fig. Number of inverse parallel relations, by protection from exchange in mass spectrometry experiments.**
(PDF)

**S3 Fig. Propensity for benign mutation by amino acid type.**
(PDF)

**S1 Table. Optimization of Fabry decision tree complexity.**
(PDF)

**S2 Table. Leading auto-ML models.**
(PDF)

**S3 Table. List of complete predictions on 203 mutations.**
(XLSX)

## Acknowledgments

The authors thank Cameron Fen for help with Auto-ML, Eric Bell for helpful discussions, and Dr. Gil Omenn for comments on the manuscript.

## Author Contributions

**Conceptualization:** Jaie Woodard, Yang Zhang.

**Funding acquisition:** Yang Zhang.

**Investigation:** Jaie Woodard, Wei Zheng.

**Methodology:** Jaie Woodard, Wei Zheng.

**Supervision:** Yang Zhang.

**Validation:** Wei Zheng.

**Writing – original draft:** Jaie Woodard.

**Writing – review & editing:** Jaie Woodard, Wei Zheng, Yang Zhang.

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
