## [Decision Letter · Decision Letter 0]

14 Jul 2021

Dear Dr. Woodard,

Thank you very much for submitting your manuscript "Protein structural features predict responsiveness to pharmacological chaperone treatment for three lysosomal storage disorders" for consideration at PLOS Computational Biology.

As with all papers reviewed by the journal, your manuscript was reviewed by members of the editorial board and by several independent reviewers. In light of the reviews (below this email), we would like to invite the resubmission of a significantly-revised version that takes into account the reviewers' comments.

We cannot make any decision about publication until we have seen the revised manuscript and your response to the reviewers' comments. Your revised manuscript is also likely to be sent to reviewers for further evaluation.

Sincerely,

Piero Fariselli

Guest Editor

PLOS Computational Biology

Nir Ben-Tal

Deputy Editor

PLOS Computational Biology

Reviewer's Responses to Questions

**Comments to the Authors:**

Reviewer #1: General

The paper: Protein structural features predict responsiveness to pharmacological chaperone treatment for three lysosomal storage disorders by Jaie Woodar et al

deserves to be published but needs changes and additions.

I found it very interesting the fact that the authors show the predictions for Gaucher disease generated using the Fabry disease or Pompe disease model separately.

The only pharmacological chaperone that has been approved for therapy is galafold (trade name). The use of pharmacological chaperones for Fabry disease has been more successful and therefore a large number of papers and data have accumulated. The authors claim to have used data from Benjamin et al (BENJAMIN, Elfrida R., et al. The validation of pharmacogenetics for the identification of Fabry patients to be treated with migalastat. Genetics in Medicine, 2017, 19.4: 430-438), but other papers also report data on many Fabry mutations and should be cited because they report exhaustive numerical data from independent groups (CITRO, Valentina, et al. The large phenotypic spectrum of Fabry disease requires graduated diagnosis and personalized therapy: A meta-analysis can help to differentiate missense mutations. International journal of molecular sciences, 2016, 17.12: 2010). It is therefore no wonder that the model derived from the data for alpha-galactosidase is more robust than that derived for alpha-glucosidase as observed by the authors themselves in the discussion (In general, based on our statistics and properties of the training data and models, we might expect the machine learning (H2O) results based on the Fabry dataset to yield the highest accuracy results) and table I . I think this point should be stressed further.

Other works have tried to predict responsive mutations for Fabry disease. In particular one paper (CUBELLIS, Maria Vittoria; BAADEN, Marc; ANDREOTTI, Giuseppina. Taming molecular flexibility to tackle rare diseases. Biochimie, 2015, 113: 54-58.)demonstrates that mutations occurring in the most flexible sites are generally more responsive. but there are exceptions in particular for glycines. It would be interesting to understand if the method used by the authors allows to explain why mutations in flexible glycines are less responsive than expected and in general the accordance of their results with those obtained using molecular dynamics.

The originality of the work consists in trying to use a model protein for which there are many in vitro experimental data for the prediction of the effect of chaperones on other less studied proteins. I would advise the authors to bolster their data by verifying what would happen if the alpha-glucosidase (Pompe disease) data were used to "predict" the data for alpha galactosidase (Fabry disease) and vice versa.

Moreover

In the Introduction

"… Pharmacological chaperones have become popular as potential treatments for several diseases, including lysosomal storage disorders, "

More correctly “some” lysosomal storage disorders

The same applies in the discussion:

"Pharmacological chaperone treatment has been explored in the case of several diseases (9, 10, 16), 317 with much of the focus centering on lysosomal storage disorders"

The trademark of the pharmacological chaperone for Fabry disease is GALAFOLD™ (migalastat): the authors should mention this

"The drug binds the protein as an inhibitor (although non-inhibitory chaperones have also been proposed, which may overcome dosing complications introduced by inhibitory effects), assisting folding in the Endoplasmic Reticulum."

References should be provided at least for the three lysosomal storage diseases that are analysed by the authors

In the Discussion

The authors make a generic comment concerning Arg to Cys mutations.

Indeed these variants have been reviewed for Fabry disease in (CITRO, Valentina, et al. The large phenotypic spectrum of Fabry disease requires graduated diagnosis and personalized therapy: A meta-analysis can help to differentiate missense mutations. International journal of molecular sciences, 2016, 17.12: 2010.)and the authors can find data in the supplementary files. R118C, R363C, R49C show different values of residual basal activity in the absence of the chaperones but are all responsive in particular R363C.

Reviewer #2: In this manuscript Woodard et al perform a structural analysis of variants within three inherited lysosomal storage disorders, comparing variants responsive to pharmacological chaperone treatment to those unresponsive to such treatment.

The article is interesting and relevant for the field of genetic diseases and precision medicine and drug discovery.

Comments:

1. Please, discuss if the the stabilization of mutant proteins can also be achieved by the substrate of the enzyme, cofactors or coenzymes.

2. In Gaucher disease, the L444P variant can be rescued by a L-idonojirimycin derivative

PMID: 28171725 and PMID: 26045184

**Have the authors made all data and (if applicable) computational code underlying the findings in their manuscript fully available?**

Reviewer #1: Yes

Reviewer #2: Yes

PLOS authors have the option to publish the peer review history of their article (what does this mean?). If published, this will include your full peer review and any attached files.

Reviewer #1: No

Reviewer #2: No
---

## [Decision Letter · Decision Letter 1]

21 Aug 2021

Dear Dr. Zhang,

We are pleased to inform you that your manuscript 'Protein structural features predict responsiveness to pharmacological chaperone treatment for three lysosomal storage disorders' has been provisionally accepted for publication in PLOS Computational Biology.

Best regards,

Piero Fariselli

Guest Editor

PLOS Computational Biology

Nir Ben-Tal

Deputy Editor

PLOS Computational Biology

Reviewer's Responses to Questions

**Comments to the Authors:**

Reviewer #1: I am satified

Reviewer #2: The authors have adressed all my concerns

**Have the authors made all data and (if applicable) computational code underlying the findings in their manuscript fully available?**

Reviewer #1: Yes

Reviewer #2: None

PLOS authors have the option to publish the peer review history of their article (what does this mean?). If published, this will include your full peer review and any attached files.

Reviewer #1: No

Reviewer #2: No

---

## [Editor Report · Acceptance letter]

9 Sep 2021

PCOMPBIOL-D-21-01060R1 

Protein structural features predict responsiveness to pharmacological chaperone treatment for three lysosomal storage disorders

Dear Dr Zhang,

I am pleased to inform you that your manuscript has been formally accepted for publication in PLOS Computational Biology. Your manuscript is now with our production department and you will be notified of the publication date in due course.

With kind regards,

Katalin Szabo
